# Affective and Dynamic Beam Search for Story Generation

**Tenghao Huang[1]**    **Ehsan Qasemi[1]**    **Bangzheng Li[1]**
**He Wang[2]**    **Faeze Brahman[3]**    **Muhao Chen[1,4]**    **Snigdha Chaturvedi[5]**
[1]University of Southern California    [2]Columbia University
[3]Allen Institute for Artificial Intelligence    [4]University of California, Davis
[5]University of North Carolina
{tenghaoh, qasemi, bangzhen}@usc.edu;
hw2687@columbia.edu;   faezeb@allenai.org;
muhchen@ucdavis.edu;   snigdha@cs.unc.edu

## Abstract

Storytelling's captivating potential makes it a fascinating research area, with implications for entertainment, education, therapy, and cognitive studies. In this paper, we propose **Aff**ective Story **Gen**erator (AFFGEN) for generating interesting narratives. AFFGEN introduces 'intriguing twists' in narratives by employing two novel techniques—Dynamic Beam Sizing and Affective Reranking. Dynamic Beam Sizing encourages less predictable, more captivating word choices using a contextual multi-arm bandit model. Affective Reranking prioritizes sentence candidates based on affect intensity. Our empirical evaluations, both automatic and human, demonstrate AFFGEN's superior performance over existing baselines in generating affectively charged and interesting narratives. Our ablation study and analysis provide insights into the strengths and weaknesses of AFFGEN.

## 1   Introduction

Stories have been a central part of human cultures for millennia, shaping societies, identities, and beliefs (Kasunic and Kaufman, 2018). However, the question of why some stories captivate us while others leave us indifferent remains intriguing. While humans can skillfully craft interesting narratives, even the most recent AI models cannot compose stories that can engage the reader for long enough. In this work, we address the task of automatically generating interesting stories.

Automatically generating interesting stories could potentially help cognitive studies by revealing patterns that make stories interesting. From an application perspective, the capability to generate interesting stories could revolutionize the fields like entertainment (Akoury et al., 2020; Thue et al., 2007), education (Zhao et al., 2022), and even therapy (Gabriel and Young, 2011).

While large language models (LLMs), such as GPT (Radford et al., 2019), have been de facto

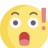
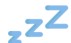

Jason is on a winter vacation at a ski resort. **He accidentally hit an avalanche trap!** He quickly grabbed his gear and rushed to the nearest shelter. However, the avalanche that followed buried him and the shelter.

**Story 1**
Interesting Narratives

Jason is on a winter vacation at a ski resort. He had been waiting for this moment all year. He was an experienced skier, and the fresh powder was perfect for some extreme skiing. Nothing could stop him now.

**Story 2**
Plain Narratives

Figure 1: Two example stories. Story 1 is an interesting story with an intriguing twist (highlighted in orange color) that was produced by AFFGEN using dynamic beam sizing. Story 2 is a relatively less interesting story with a straightforward and predictable plot.

winners in generating coherent text, their prowess in creating narratives that captivate human interest leaves much to be desired. LLMs' coherence is mainly rooted in their training objective that incentivizes text likelihood which is not necessarily correlated with human quality judgements (Holtzman et al., 2019; Zhang et al., 2021) or writing style (Gehrmann et al., 2019). The concept of "interesting stories" is also highly subjective and context-dependent (Roemmele, 2021). Previous research in the field increases "interest" in the story by structural planning to control specific aspects of the story, e.g. modeling the emotional flow of the protagonist (Luo et al., 2019; Brahman and Chaturvedi, 2020) or incorporating flashbacks (Han et al., 2022). However, such methods ignore that text complexity and quality also raise its interestingness (Schraw et al., 2001).

Bradley and Lang (1999) advocates for decorating the plot with affective terms to increase the suspense and intensity of the story that results in control of the audience's emotions (Delatorre et al., 2016). With this motivation, we propose **Aff**ective Story **Gen**erator (AFFGEN)[1] that con-

---

[1]Code is available at https://github.com/tenghaohuang/AFFGEN.git

trols text coherence and leverages words' affective dimensions to promote text interestingness. Our method is based on two key ideas. First, in beam-search-based decoding of language models, occasionally exploring larger beams can help in generating slightly lower probability but potentially more interesting words. Second, switching between large and small beams can help in maintaining the balance between coherence and interestingness. We use these ideas to generate stories with an **intriguing twist**. Figure 1 shows an example of an interesting story, Story 1, with an intriguing twist (highlighted in orange color) that was produced by dynamically using different beam sizes. It also shows an uninteresting story, Story 2, that used a comparable language model but with a constant beam size.

To generate an interesting story, AFFGEN first identifies where to generate the intriguing twist that would push the story to be more interesting. Then it generates the intriguing twist using two novel techniques, i.e. *Dynamic Beam Sizing* and *Affective Reranking*. In dynamic beam sizing, AFFGEN uses a contextual bandit model (Thompson, 1933) to dynamically explore different beam sizes thus encouraging the model to select words that are less predictable and more intriguing without compromising coherence. In affective reranking, AFFGEN reranks possible candidates for the sentence to be generated according to their arousal and valence scores (Mohammad, 2018), thereby modulating the emotional dynamics of the story.

Our automatic and human evaluations show that stories generated by AFFGEN are more engaging than the baselines without sacrificing coherence. Our ablation studies and analysis provide deeper insights into the functioning of AFFGEN

Our contributions are:

- We propose the task of generating interesting stories.

- We propose AFFGEN, a language model that uses a novel contextual bandit-based decoding algorithm and explores dynamic beam sizes and affective reranking.

- We conduct automatic and human evaluations to empirically demonstrate that AFFGEN can produce interesting and coherent narratives.

- We conduct ablation studies and analysis to further understand the working of AFFGEN.

## 2 Related Works

We discuss two lines of related work that are closely relevant to this study.

**Story Generation.** Early research on story generation explored symbolic planning methods (PÉrez and Sharples, 2001; Porteous and Cavazza, 2009; Riedl and Young, 2010) that used predefined rules and structures to generate stories. Later efforts used neural methods (Jain et al., 2017; Peng et al., 2018; Fan et al., 2018; Puduppully et al., 2019; Zhai et al., 2019; Yao et al., 2019; Wang et al., 2021; Peng et al., 2022).

However, generating interesting stories has remained a challenge due to the subjective nature of "interestingness" (Roemmele, 2021). Some previous work has attempted to generate interesting stories by controlling specific aspects of the generated content, such as modeling emotions (Luo et al., 2019; Brahman and Chaturvedi, 2020), flashbacks (Han et al., 2022), personas (Zhang et al., 2022), topics (Lin and Riedl, 2021), and social relationships (Vijjini et al., 2022). Alhussain and Azmi (2021) pointed out factors that could lead to interesting narratives, such as suspense (Tan and Fasting, 1996), discourse (Genette, 1980), and characters (Liu et al., 2020). This work differs from these approaches in the sense that it focuses on generating interesting content by choosing more affective, and not necessarily high-likelihood, words.

**Sampling strategies for decoding.** One of the commonly used strategies in neural text (and story) generation is Nucleus Sampling (Holtzman et al., 2019). This method involves selecting a subset of the vocabulary, called the nucleus, from which the next word is sampled. Another strategy is Top-$k$ Sampling (Fan et al., 2018), which only considers the $k$ most probable words for the next word. Meister et al. (2023) proposed an information-theoretic strategy, *Locally Typical* Sampling, with the aim of making the model's output more human-like.

Our approach differs from these existing strategies in two key perspectives. First, while previous works primarily aim to encourage generation fluency and diversity we focus on including more affective terms during decoding. Second, we use re-scoring, which involves adjusting the probabilities of the words based on additional criteria, rather than solely relying on the logits distribution generated by the model. This allows us to further enhance the diversity and affective quality of the

generated text.

## 3 Problem statement

Given a sentence, $\mathbf{s_1}$, as a prompt that represents the first sentence of a story, our goal is to generate an interesting story represented as a sequence of generated sentences $\mathbf{s_2}, \mathbf{s_3}, \ldots, \mathbf{s_N}$. Each sentence is a sequence of tokens. In this paper, one of these generated sentences serves as the intriguing twist in the narrative.

## 4 Controlled Affective Story Generator

This section presents the Controlled Affective Story Generator (AFFGEN), a narrative generation model designed to produce interesting stories. AFFGEN operates in two key stages. First, it identifies the position of the sentence that should contain the intriguing twist, $p_{IT}$ (§4.1). Then, it generates the story in the left-to-right manner using a language model. For generating sentences that do not contain the intriguing twist, it uses a standard decoding algorithm since the focus is on maintaining narrative coherence (§4.2). For generating the sentence that contains the intriguing twist, it uses our proposed decoding algorithm based on Dynamic Beam Sizing and Affective Reranking since the focus is on balancing emotional arousal, interestingness, and coherence (§4.3).

### 4.1 Position of the intriguing twist

Narratives are highly structured texts. Freytag's pyramid (Freytag, 1908), a widely recognized model of narrative structure, delineates the story into five key components: exposition, rising action, climax, falling action, and resolution.

Given the prompt sentence, $\mathbf{s_1}$, our objective is to determine the most suitable location for the climax or the intriguing twist, $n_{IT} \in \{2, 3, \ldots N\}$. There has been some work on identifying the climax or turning point in a given story (Ouyang and McKeown, 2015; Wang et al., 2022; Vijayaraghavan and Roy, 2023). We employ a data-driven approach inspired by the work of Wilmot and Keller (2020). Their methodology operates on the premise that if the embedding of two sentences is sufficiently distant, the latter sentence can be deemed unexpected or interesting with respect to the former sentence. They use this idea to identify the sentence that presents the turning point or intriguing twist in a narrative.

Our data-driven approach utilizes the Writing-Prompts dataset (Fan et al., 2018), a collection of human-written stories. We use this dataset to form a distribution, $D(n)$, which corresponds to the probability of observing the intriguing twist at the $n^{th}$ sentence. During inference, AFFGEN samples a relative position $n_{IT}$ from $D(n)$ to pinpoint the location of the sentence that would be the intriguing twist in the story that will be generated

$$n_{IT} \sim D(n).$$

Next, we discuss how AFFGEN generates the various sentences of the story.

### 4.2 Base Storyteller

For generating sentences that do not contain an intriguing twist ($\mathbf{s_i}$'s $\forall i \notin \{1, n_{IT}\}$), the focus is on maintaining narrative coherence. We use a GPT-based language model (Radford et al., 2019; Brown et al., 2020) which has shown promising performance on story generation (Brahman and Chaturvedi, 2020; Clark and Smith, 2021). We fine-tune the language model on a dataset of stories (§5.1) by minimizing the negative conditional log-likelihood:

$$NLL = -\log \prod_{i=1}^{n} p(w_i | w_1, ..., w_{i-1}). \quad (1)$$

where $w_i$'s represents the tokens of the story. We use beam search for inference in this model.

### 4.3 Generating Intriguing Twist

To generate the sentence that contains the intriguing twist in the narrative, $\mathbf{s}_{IT}$, we use the fine-tuned language model from §4.2 but with a novel beam search-based decoding. Our decoding method uses Dynamic Beam Sizing and Affective Reranking to produce interesting text.

**Dynamic Beam Sizing.** The motivation behind our beam search-based decoding algorithm is that while a small beam size helps in producing coherent text, by expanding the beam size of the PLM, we can explore slightly lower probability but potentially more intriguing words. However, maintaining a large beam size throughout is also not desirable because not all words in a sentence need to be interesting. A large beam throughout can also slow down the inference process and require more resources. So during inference, the model needs to dynamically switch between large and small beam

sizes to balance the tradeoff between the coherence and interestingness of the generated text.

To address this, we introduce Dynamic Beam Sizing, where depending on the context, the model decides the beam size before generating a token. For practical purposes, we assume that the beam size can take one of $k$ values $\{b^1, b^2...b^k\}$, and the model has to choose one. We cast the problem of choosing a beam size as a contextual $k$-arm bandit problem (Langford and Zhang, 2007), where the *arms* of the bandit are the various beam sizes. The bandit's choice of beam size at time step or *trial*, $t$, depends on the the *context* of the bandit. The *context* considers the tokens generated so far for the intriguing twist sentence, $\mathbf{s}_{IT}$. We use $\mathbf{s_{IT,t-1}}$ to refer to the sequence of tokens in this partial sentence and represent the *context* using following features:

1. Arousal score: The arousal score of the sentence generated so far, $\mathbf{s_{IT,t-1}}$. The arousal score of a partial sentence, viewed as a sequence of tokens, $\mathbf{s}$, of length $n$ is:

$$A(\mathbf{s}) = \sum_{i=1}^{n} a(w_i) \qquad (2)$$

where $a(w_i)$ is the arousal score of the $i^{th}$ token obtained from the NRC Word-Emotion Association lexicon (Mohammad, 2018). Since longer sentences can accumulate higher arousal scores, we divide the arousal score by a length normalizing factor (Wu et al., 2016). The length normalizing factor for a sentence of length $n$ is:

$$lp(n) = \frac{(5+n)^\lambda}{(5+1)^\lambda} \qquad (3)$$

where $\lambda$ is the normalization coefficient.

2. Event trigger likelihood: Sims et al. (2019) points out that in narratives there are certain words in a sentence that trigger interesting literary events. E.g. In the sentence "... Stephen leaned his arms on ...". The word "leaned" is an event trigger. Identifying such event triggers can help in locating the interesting part of a sentence, which in turn will help in deciding whether to choose a larger beam. With this motivation, we train a RoBERTa (Liu et al., 2019) based predictor that given a partial sentence predicts whether the next token would be the trigger for an interesting literary event. We provide the partial sentence generated so far, $\mathbf{s_{IT,t-1}}$, as the input to this predictor and use the likelihood assigned by it (for the next token to be an event

trigger) as a feature.

3. Sequence length: Length of the partial sentence generated so far, $\mathbf{s_{IT,t-1}}$. Knowing where the model is, in terms of position, can help it decide whether to generate an interesting token next.

4. Perplexity: The model's perplexity on the partial sentence generated so far, $\mathbf{s_{IT,t-1}}$. This helps in maintaining coherence.

For choosing an *arm* $b \in \{b^1, b^2...b^k\}$, the bandit also receives a *payoff*. The *payoff* accounts for all the candidate sequences in the beam $\{\mathbf{c}^1, \mathbf{c}^2, ..., \mathbf{c}^b\}$. Each $\mathbf{c}^i$ is basically a concatenation of the partial sentence generated so far, $\mathbf{s}_{IT,t-1}$, and the $i^{th}$ token in the beam. The *payoff* rewards beams that contain candidate sequences with high arousal scores (to promote interestingness) and low perplexity (to promote coherence). It also penalizes large beam sizes to encourage using fewer compute resources. Mathematically, the payoff value $R(b_t, t)$, for choosing a beam size, $b_t$, at time step $t$, is defined as:

$$R(b_t, t) = \max_{i \in [1, b_t]} (A(\mathbf{c}^i) - \alpha \cdot \text{ppl}(\mathbf{c}^i) - \beta \cdot |b_t|), \quad (4)$$

where $\alpha, \beta$ are coefficients for each component, $A(\mathbf{c})$ and $\text{ppl}(\mathbf{c})$ represent the arousal score (as defined in Eqn. 2) and the perplexity of the candidate sequence $\mathbf{c}$ respectively, and $|b_t|$ represents the size of beam $b_t$.

Given the set of $k$ choices for beam sizes $\{b^1, b^2, \ldots b^k\}$, the optimal beam size $b_t^*$ at timestep $t$ is given by

$$b_t^* = \underset{i \in [1,k]}{\text{argmax}} \ R(b_t^i, t) \qquad (5)$$

Correspondingly, the optimal payoff at time step, $t$ is $R(b_t^*, t)$.

Using the LinUCB (Upper Confidence Bound) algorithm (Li et al., 2010), we optimize the bandit model by minimizing regret $L$ defined as:

$$L = \mathbb{E}[\Sigma_{t=1}^{T} R(b_t^*, t)] - \mathbb{E}[\Sigma_{t=1}^{T} R(b_t, t)] \qquad (6)$$

where $T$ is the total number of time steps or the total number of tokens in $\mathbf{s}_{IT}$.

**Affective Reranking.** While Dynamic Beam Sizing introduces more arousing content, it does not consider the variation of emotions associated with the content. Chung et al. (2022) highlighted that variation of emotional arc (Reagan et al., 2016) can make a story more engaging. We, therefore, introduce Affective Reranking.

Let $\{\mathbf{s_{IT}}^1, \mathbf{s_{IT}}^2 \ldots \mathbf{s_{IT}}^b\}$ be the candidate sentences that are generated as potential intriguing twists in the beam. The best candidate should have a high arousal score and should also have high affective contrast. We quantify affective contrast as the difference in the valence scores of the candidate sentence and the story generated so far. Valence score of a sequence of tokens, $v$, is the length-normalized cumulative valence score of its individual tokens. We use the NRC-VAD lexicon (Mohammad, 2018) to obtain valence scores of tokens.

We select the best candidate for the intriguing twist sentence $\mathbf{s}^*$ such that:

$$\mathbf{s}^* = \underset{i \in [1,b]}{\arg\max} \; A(\mathbf{s_{IT}}^i) + |v(\mathbf{s_{IT}}^i) - v(\mathbf{s_{1:IT-1}})| \quad (7)$$

## 5 Empirical Evaluation

In this section, we describe our experiments.

### 5.1 Experimental Setup

**Dataset.** For our experiments, we use the ROC-Stories dataset (Mostafazadeh et al., 2016), a large collection of 100k five-sentence 'commonsense' stories about everyday events. We held out 1k stories each for validation and testing and use the first sentence of every story as the prompt. We chose this dataset because it allows us to assess the performance of our model's ability to learn from a collection of everyday life stories and improvise them to be interesting. The short nature of these stories also makes the manual assessment of narrative quality feasible during human evaluation which otherwise would have been difficult. This focus on short stories, however, does not limit the potential application of our model to longer narratives.

Our base storyteller is trained on the ROCStories dataset. The contextual bandit model is trained in an unsupervised manner, relying on the internal regret function.

**Implementation Details.** All hyperparameters were set based on the performance on the validation set. We used $\alpha = 0.00015$, $\beta = 0.0003$ in Eqn. 4 and $\lambda = 1.5$ in Eqn. 3. We trained the bandit model on single A5000 for 10 epochs and it chose between three beam sizes of 10, 30 and 60.

**Baselines.** Our primary baseline is GPT2 fine-tuned on the RocStories dataset since it is widely recognized for its story generation capabilities (Brahman and Chaturvedi, 2020). We use GPT3 as

| Model | PPL ↓ | Uni ↑ | RUB ↑ | Aro ↑ |
|---|---|---|---|---|
| GPT2 | 26.77 | 0.021 | 0.1546 | 0.45 |
| AFFGEN-2 | 40.27 | 0.019 | **0.1556** | 0.51 |
| GPT3 | **18.90**\* | 0.028 | 0.1541 | 0.46 |
| AFFGEN-3 | 25.66 | **0.029** | 0.1547 | **0.53**\* |

Table 1: Automatic evaluation of AFFGEN using Perplexity (PPL), UNION score (Uni) (Guan and Huang, 2020), (RUB) score (Tao et al., 2018), and Arousal score (Aro). ↑ and ↓ indicate if higher or lower scores are desirable. Bold fonts indicate best scores and \* indicates statistical significance ($p < 0.01$). The results indicate that both versions of AFFGEN can generate interesting stories without compromising coherence.

a baseline to compare with a large language model. For GPT3, we use the following prompt [2] (after experimentation): "Continue writing an interesting story using the following context, <context>. The total length of the story should be five sentences. The total words limit is 60 words.""

### 5.2 Automatic Evaluation

Table 1 presents a comparison between AFFGEN and baseline methods. We use two versions of our model, AFFGEN-2 and AFFGEN-3. They use fine-tuned GPT-2 and GPT-3 as the base storytellers (§4.2). We observe that both versions of AFFGEN have higher perplexity (PPL) scores than the baselines. This, however, is expected and does not imply low coherence because AFFGEN encourages using low-likelihood words during the decoding process to generate interesting content.

For a better evaluation of coherence, we consider the UNION (UNI) (Guan and Huang, 2020) and RUBER (RUB) scores (Tao et al., 2018). UNION is a reference-free score specially designed for evaluating open-ended story generation models. RUBER is a hybrid of referenced and unreferenced metric used for evaluating dialog systems. We only use its unreferenced part to evaluate the quality of a piece of text (story) generated in response to a query (the story prompt). A higher value for these scores is better. We observe that for these scores versions of AFFGEN either perform better than or comparable to the baselines. This indicates that AFFGEN is capable of generating coherent narratives.

For evaluating how interesting the stories are, we measure their per-token Arousal score (Aro) (Eqn. 2) which quantifies their affect level. A higher value is better for this score. We observe that

---

[2]Please refer to Table 7 for more pormpt details.

| Evaluation Criteria | Win | Lose | Tie |
|---|---|---|---|
| Coherence | **50.5**\* | 40.7 | 8.8 |
| Emotional Engagement | **53.0**\* | 40.3 | 6.7 |
| Empathy | **53.8**\* | 40.2 | 6.0 |
| Interestingness | **54.9**\* | 41.1 | 4.0 |
| Overall Preference | **52.7**\* | 39.6 | 7.7 |

Table 2: Human evaluation of AFFGEN vs GPT-3. AFFGEN generates better stories across all measures. \* indicates statistical significance (p<0.1 for coherence and p<0.05 for others).

| Evaluation Criteria | Win | Lose | Tie |
|---|---|---|---|
| Coherence | 14.5 | **38.8**\* | 46.7 |
| Emotional Engagement | **55.5**\* | 24.8 | 19.7 |
| Empathy | **40.8**\* | 28.6 | 30.6 |
| Interestingness | **45.3**\* | 26.3 | 28.4 |
| Overall Preference | **45.3** | 35.7 | 19.0 |

Table 3: Human evaluation of AFFGEN vs ChatGPT. AFFGEN generates not as coherent but more interesting and empathetic stories. \* indicates statistical significance (p<0.05).

both versions of AFFGEN outperform the baselines with AFFGEN-3 achieving the highest score. This indicates that AFFGEN generates more interesting stories.

## 5.3 Human Evaluation

In order to assess the performance of AFFGEN, a comprehensive human evaluation was conducted on the Amazon Mechanical Turk (AMT) platform. A total of 100 instances were randomly selected from our test set. We feed their initial sentences as prompts for generating stories using AFFGEN-3 and GPT-3, our stronger baseline. To eliminate any potential bias, the presentation order of the two stories was randomized. The Turkers then selected the better of the stories according to 6 criteria: coherence, emotional engagement, empathy, interestingness, and overall preference. These criteria were chosen based on prior research conducted by Chhun et al. (2022). The Turkers could also select an *"equally good"* option. The Turkers were explicitly instructed to solely consider the given criterion when evaluating, except when expressing an overall preference. In the appendix, Figure 4 showcases a screenshot of our AMT setup. We specifically utilized Master annotators predominantly from English-speaking countries (US, UK, Canada, and Australia). We evaluated 200 stories in total, and each pair was assessed by three different annotators. We discuss the results shown in Table 2 below. All differences in this table are statistically significant (p<0.1 for coherence and p<0.05 for others) and the inter-annotator agreement is 0.58 (moderate agreement).

**Coherence** evaluated the logical flow and connection between the different elements of the story. For this criterion, judges found stories generated by AFFGEN-3 to be more coherent than those generated by GPT-3 in 50.5% of instances, while AFFGEN-3's stories were considered less coherent in 40.7% of cases. The remaining 8.8% resulted

in a tie. This indicates that AFFGEN does not compromise on coherence while generating stories.

**Emotional Engagement** evaluated how effectively a story conveys a range and intensity of emotions that capture and hold the reader's attention and create a sense of emotional depth and complexity. For this criterion, judges found stories generated by AFFGEN-3 to be more emotionally engaging than GPT-3 in 53.0% and less emotionally engaging in 40.3% of the cases. This demonstrates AFFGEN's stronger ability to evoke emotions in readers.

**Empathy** evaluated if the story arouses the readers' empathy for the characters. The conflicts and challenges described in stories can create situations that make the readers project their own emotions and thoughts onto the characters, keeping them invested and engaged. For this criterion, AFFGEN-3 outperformed GPT-3 by a large gap of 13.6% (53.8% wins and 40.2% losses). This demonstrates that AFFGEN can generate emotionally resonant content.

**Interestingness** evaluates the story's ability to be compelling and engaging. For this criterion also, AFFGEN-3 outperformed GPT-3 by a large gap of 13.8% (54.9% wins and 41.1% losses). This demonstrates AFFGEN's its superiority in keeping the reader's interest while generating stories.

**Overall Preference** Finally, we observed that overall, the judges preferred AFFGEN over the baseline in 52.7% of the cases (as compared to preferring baseline over AFFGEN in 39.6% cases).

To conclude, the human evaluation results provide strong evidence of the superiority of the AFFGEN in various critical aspects of open-ended story generation underlying its ability to generate interesting and engaging stories while maintaining coherence.

|                    | UNION  | RUBER  | Arousal |
|--------------------|--------|--------|---------|
| AFFGEN$_{10}$      | -0.007 | -0.012 | -0.018  |
| AFFGEN$_{30}$      | -0.002 | -0.007 | 0.024   |
| AFFGEN$_{60}$      | -0.005 | -0.006 | 0.047   |
| AFFGEN $- AR$      | 0.002  | -0.001 | -0.070  |

Table 4: Performance of ablated versions of AFFGEN with static beam sizes relative to AFFGEN. Subscripts indicate the beam sizes. A negative score indicates that the ablated version did not perform as well as AFFGEN. These results indicate that it is important to explore large beam sizes in a dynamic manner to generate interesting and coherent stories.

## 5.4 Comparison with ChatGPT

In this section we compare AFFGEN with a large language model, ChatGPT [3]. We used a human evaluation setup similar to that described in §5.3. These annotations were performed by expert annotators who were students of literature theories. For generating stories with ChatGPT, we experimented with different prompts and the final prompt is shown in Table 7. Table 3 shows the results. Annotators expectedly found ChatGPT's stories to be more coherent. Our initial analysis also revealed ChatGPT text to have more sophisticated structure. However, annotators found AFFGEN's stories to be significantly more emphathy-evoking and interesting. Because of this, the annotators preferred AFFGEN over ChatGPT in the overall preference.

## 5.5 Ablation Study

We now describe our ablation study in which we investigate the importance of exploring different beam sizes and affective reranking. In our experiments reported so far, we made AFFGEN explore three different beam sizes during decoding. In this study, we design ablated versions of AFFGEN that only uses one of the three beam sizes. We call them AFFGEN$_{10}$, AFFGEN$_{30}$, and AFFGEN$_{60}$, where the subscript indicates the beam size being used. The first three rows of Table 4 reports the relative performance of these versions with AFFGEN. All models use fine-tuned GPT-2 as the base storyteller. For all scores, a negative score indicates that the ablated version did not perform as well as AFFGEN (and vise versa). We can see that for most of the ablated versions, the UNION and RUBER scores are negative. This means that the stories generated by the ablated versions are less coherent than the full model. In terms of Arousal scores, AFFGEN$_{10}$

[3]OpenAI. (2023). ChatGPT (May 24th version) [Large language model]. https://chat.openai.com

produces less arousing stories than AFFGEN but AFFGEN$_{30}$, and AFFGEN$_{60}$ produce more arousing stories than AFFGEN. This aligns with our initial intuition that a larger beam size helps the model generate more interesting content. However, because of large but static beam sizes, the stories generated by these two versions were less coherent than those generated by AFFGEN.

Next, we also consider another version of AFFGEN but without Affective Reranking. The relative performance of this model is shown in the last row of Table 4. We can see that the performance of this version is quite close to the baseline. Also, while its coherence is comparable to AFFGEN, the arousal score is particularly worse indicating the importance of this component in generating interesting content.

Overall, we can draw two conclusions from this ablation study. First, exploring large beam sizes and affective reranking can help in generating more interesting content. Second, it is important to dynamically switch between larger and smaller beams to balance interestingness and coherence.

## 5.6 Expansion to Longer Narratives

Our experiments have used RocStories which are short in nature. This focus on short stories does not limit the potential application of our model to longer narratives. Jolles (2017) points out that stories could be condensed into "simple forms". Story composition could be viewed as a process of expanding these simple forms into presentable longer narratives. Table 10 presents expanded AFFGEN-generated stories and compared with vanilla ChatGPT generated stories. With the help of the five-sentence interesting plots produced by AFFGEN, ChatGPT expand them into better stories comparing to vanilla ChatGPT generated stories.

## 5.7 Dynamic Beam Sizing

We now investigate how the beam size changes as AFFGEN generates an interesting sentence. Figure 2 shows the average beam size used to generate at different positions of a typical sentence. We observe that AFFGEN is using larger beam sizes for the first few tokens. Our manual analysis revealed that the interesting words indeed appear earlier in a sentence, in general.

Since the model is capable of transitioning between beam sizes, we plot a heat map of the transitions shown in Figure 3. Each cell shows the probability of transitioning from beam size on Y

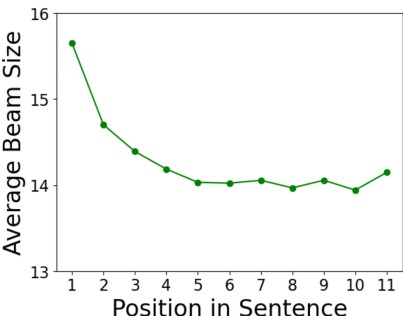

Figure 2: Average beam size used to generate at different positions of a sentence.

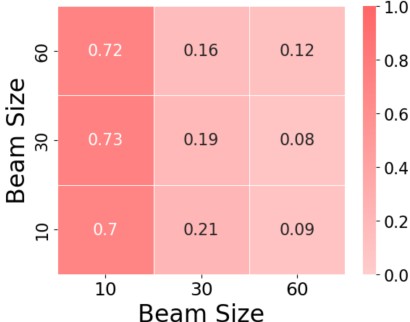

Figure 3: Transitional probability between beam sizes.

axis to a beam size on the X axis. Darker color indicates higher probabilities. We observe that in general, while AFFGEN has a tendency to stick to a chosen beam size ($\sim 70\%$), it does transition to different beam sizes about $30\%$ of the times indicating the importance of switching between beam sizes.

## 5.8 Qualitative Analysis

During the human evaluation (§5.3), when asking for preferences for the stories, we also asked the judges to provide explanations for their choices. We then analyzed the explanations to further analyze the stories generated by AFFGEN. Table 5 show an example of stories generated by GPT-3 and AFFGEN for the same input prompt as well as the human-provided explanation. While both stories have a happy ending, AFFGEN's story introduces a plot complication, where the protagonist, Grayson's, initial attempt to bake a cake fails. He resolves the situation through determined efforts, creating a narrative of perseverance. Compared to the baseline, the plot in AFFGEN's story becomes more complicated and has more ups and downs, which enhances the emotional engagement and interest of the reader. The AMT judges noted that the AFFGEN's story was more emotionally expressive.

Table 8 in the Appendix provides more comparative examples of stories generated by the baseline and AFFGEN and corresponding explanations. Analyzing explanations for story pairs we found that the judges preferred AFFGEN's stories because they presented a shift in mood enhancing the affect they had on the reader. AFFGEN's stories also presented unexpected twists which provide a relief from the story's prevalent theme and increases its interest. In contract, the baseline stories were banal and conflict-less. Sometimes AFFGEN's stories introduced a melancholic theme, but the judges still found them pleasant. This aligns with the narratological theory presented by Massumi (2002) who argues that there is a gap between the content and the effect on the receiver's end. As a result, audience often rate "sad" scenes in the film as the "most pleasant". Overall, AFFGEN was found to be better at generating more emotionally captivating and interesting stories leading to to better storytelling experience.

## 5.9 Error Analysis

Using the judges' explanations provided during the human evaluation, we also conduct an error analysis to identify issues encountered during story generation by AFFGEN. Table 9 in Appendix shows some examples of story pairs in which the judges did not prefer AFFGEN's stories over GPT-3's stories and their explanations. We observe that while AFFGEN introduces an intriguing twist in the story, it sometimes suffers from typical language modeling challenges like repetitive phrases and ideas (Story 2) and incoherence. Often the incoherence is caused by a lack of commonsense knowledge like sunglasses cannot change eye colors (Story 1), and if a toy breaks, it cannot function (Story 3). This aligns with the proposition made by Alhussain and Azmi (2021) that coherence (and also causality) are fundamental in storytelling. Without them, the story may disintegrate into inconsistent fragments.

## 6 Conclusion

This paper addresses the task of generating interesting stories. For this, we present, AFFGEN, a language model that uses a novel contextual bandit-based decoding mechanism. This new decoding mechanism enables the AFFGEN to dynamically explore different beam sizes and rerank based on affective quality of the text. Our experiments indicate that AFFGEN can generate interesting but

| | |
|---|---|
| GPT-3 | Grayson wanted to bake his brother a birthday cake, so he got out his mom's recipe book and started searching for the perfect one. He decided to go all out and make a four layer chocolate cake with cream cheese frosting. His brother was so impressed that he wolfed down 2 pieces in a row. |
| AFFGEN | Grayson wanted to bake his brother a birthday cake. **The first attempt turned out terribly bad and ruined.** Grayson was determined to make the perfect cake for his brother's birthday. Undeterred, Grayson started again from scratch and was finally able to make a delicious cake that his brother loved. |
| Human Explanation | The story by AFFGEN is more emotionally expressive |

Table 5: Sample stories generated by GPT-3 and AFFGEN. As indicated by the judge, AFFGEN's story is more emotionally engaging.

coherent narratives. Our ablation studies underscore the importance of dynamic beam sizing and affective reranking, and our qualitative and error analysis point to the strengths and weaknesses of our model. We hope that this ability to compose interesting narratives can open new dimensions of computational creativity, driving the generation of unique and captivating content at an unprecedented scale and speed.

## Acknowledgement

We appreciate the reviewers for their insightful comments and suggestions. Tenghao Huang and Muhao Chen were supported by the NSF Grant IIS 2105329, an Amazon Research Award and a Keston Exploratory Research Award. Ehsan Qasemi was supported by the DARPA MCS program under Contract No.N660011924033 with the United States Office Of Naval Research. Computing of this work was partly supported by a subaward of NSF Cloudbank 1925001 through UCSD.

## Limitations

Our study has the following limitations.

We assume a single sentence containing an intriguing twist can enhance a story's interestingness. This paper focused on how to generate that intriguing twist. However, a story can potentially benefit from multiple interesting sentences and future works can investigate into how frequently and where to generate interesting content.

We adopted a simple data-driven approach for deciding where to put the sentence that contains the intriguing twist. It samples from a distribution learned from a collection of stories. Future work could work on more sophisticated methods that consider the preceding narrative context for deciding when to describe an interesting twist so that it integrates better with the story being generated.

For practical purposes, the bandit model discretized beam sizes. However, beam size is a continuous variable, and discretizing it can restrict the

model from exploring all possible values.

Our experiments used GPT-2 and GPT-3 as the base storyteller for generating the stories. However, we see AFFGEN as a framework that could incorporate other language models and future work can investigate this aspect.

Our experiments explored short and fictional narratives. Future work could investigate advanced planning and strategies for composing longer stories or non-fictional content.

Our dataset and experiments use only one language - English. We did not investigate the model's capabilities to generate stories in other languages.

## Ethical considerations

Our experiments use a publicly available dataset. Previous work (Huang et al., 2021) has shown that it contains gender-related biases and storytelling models that use this dataset can replicate and amplify these biases. Our model also encourages low-perplexity text, which could unintentionally encourage biased, violent, or sexually explicit content. Since we have not employed any bias or toxicity removal methods, applications of our work should control for inappropriate content.

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

## A HIT Example

Figure 4 illustrates a screenshot of the form presented to the annotators on the AMT.

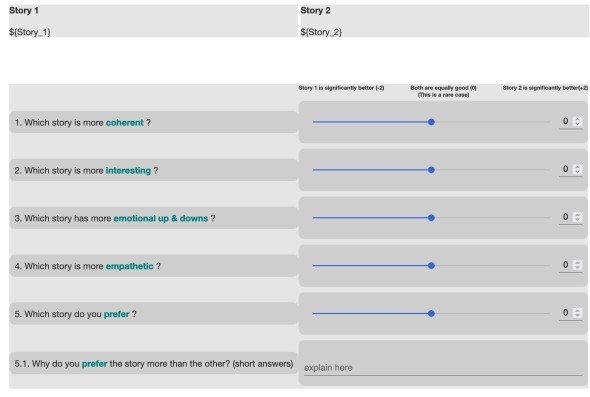

Figure 4: Screenshot of the AMT survey used for human evaluations in §5.3

## B Prompts for LLM

In our exploration, we investigated multiple prompt optimization techniques. Initially, we utilized demonstrations as a potential avenue. Yet, as illustrated in Table Tab. 6, it became evident that the resultant narrative exhibited a significant degree of repetitiveness with the given demonstration. An attempt to introduce diversity by leveraging a varied ensemble of demonstrations did not yield improvements in reducing narrative redundancy. Subsequently, we explored the incorporation of specific keywords within prompts to guide the generative process. For instance, by prompting LLM "Write a five-sentence interesting story with mystery", we observed that while such specificity could yield captivating narratives, they bore a recurring thematic redundancy. To exemplify, from a sample of ten stories generated using the "mystery-enhanced" prompt, it was observed that seven out of ten generated stories converged around the theme of a hidden secret being revealed towards the conclusion. Representative narratives included scenarios such as a photographer unveiling a mystery within a haunted lighthouse, a young woman unearthing a concealed garden, and an individual discerning a hidden cache of treasure, etc. In consideration of these findings, and without loss of generality, we opted for the current prompt for subsequent generations.

## C Stories with Annotator Preferences

Table 8 shows more sample stories generated by GPT-3 and AFFGEN. We present detailed qualita-

| Speaker | Content |
|---|---|
| User | Here is an interesting story. "Melissa invited Mark to the park for a picnic. The weather turned bad when thunder hit. Mark and Melissa decided to wait out the storm, and when the rain cleared, they had the most romantic picnic they'd ever experienced, watching the sun set over the park". Write an interesting story based on the context "Tom is going to fish on Saturday". The story should be five-sentence long and 60 words maximum. |
| ChatGPT | Tom eagerly awaited Saturday, planning a day of fishing at his beloved lakeside spot. As dawn broke, he cast his line, lost in the tranquility. Suddenly, dark clouds loomed, and rain poured down. Seeking refuge under a tree, he met Sarah, another angler. The rain ceased, but Tom left with a newfound friend and a heartwarming memory. |

Table 6: Attempt to optimize prompts through demonstration

| Task | Prompts for LLM |
|---|---|
| GPT-3 infilling | "Continue writing a story using the following context, <context>. The total length of the story should be <n> sentences. The total words limit is <n*12> words." |
| GPT-3 | "Continue writing an interesting story using the following context, <context>. The total length of the story should be <n> sentences. The total words limit is 60 words." |
| ChatGPT | "Write a five-sentence interesting short story using the following sentence as the first sentence: <context>." |

Table 7: Prompts for Large Language Models

tive analysis in §5.8.

## D Error Analysis

Table 9 shows some examples of story pairs in which the judges did not prefer AFFGEN's stories over GPT-3's stories and their explanations. We present detailed qualitative analysis in §5.9.

| ID | Baseline | AFFGEN story | Explanation |
|---|---|---|---|
| 1 | Melissa invited Mark to the park for a picnic. They had a great time, playing games and eating the delicious food she had prepared. They laughed, talked and made plans for their future. It was a perfect day that they would remember forever. | Melissa invited Mark to the park for a picnic. **The weather turned bad when thunder hit.** Mark and Melissa decided to wait out the storm, and when the rain cleared, they had the most romantic picnic they'd ever experienced, watching the sun set over the park. | <AFFGEN's> story is more romantic. |
| 2 | Ella wrote her grandma a letter, filled with stories about her first day at school. She told her about the new friends she made, the lunch she had, and the fun she had learning. Grandma was so proud of her, and Ella couldn't wait to get a letter back. | Ella wrote her grandma a letter. Ella's grandma wrote back with an unexpected news. **Her favorite aunt died unexpectedly of cancer!** Ella was saddened to hear the news, but was also comforted in knowing that her aunt had lived a full and happy life. | Emotional condolences gives <AFFGEN's> story an interesting and sad theme |
| 3 | Jen has a big test tomorrow and she's determined to ace it. She spends hours studying, memorizing facts and theories, and rehearsing for the exam. She knows she's ready and is confident that she'll get the grade she desires. | Jen has a big test tomorrow. Jen stayed up all night studying, but still felt unprepared. **Her roommate surprised and excited with good news!** She had found out that the professor had decided to give everyone an extra day to study for the test. | The <Baseline> story has no bumps, it's so banal |

Table 8: More sample stories generated by GPT-3 and AFFGEN. As indicated by the judge, AFFGEN's story is more preferred by annotators in various aspects.

| ID | Baseline | AFFGEN story | Explanation |
|---|---|---|---|
| 1 | Hazel wished for blue eyes every night, but her green gaze never changed. She felt like a different person with blue eyes; she was sure they'd bring her the confidence she craved. One day, she found a mysterious pair of blue contacts. She put them on, and it was like a whole new world opened up for her. | Hazel's eyes were green, but she really wanted blue eyes. **The school offered special free sunglasses!** Hazel was always envious of her friends' blue eyes. With the school's offer of free sunglasses, she was determined to make her dream come true. She put on the sunglasses and instantly felt like she had the most beautiful blue eyes in the world. | The <Baseline> story is interesting |
| 2 | We went out last night, dancing and laughing the night away. I had the time of my life, and I felt like nothing could stop us. I never wanted the night to end, and I will never forget it. | We went out last night. **My girlfriend was excited to meet new friends!** We had a great time, and my girlfriend made a few new friends. We stayed out late, and although I was tired, I was happy to see her having a good time. We both agreed it was a great night out and we can't wait to do it again. | The <AFFGEN > story has too much repetitions and is basically dull, the first one sounds very authentic |
| 3 | Joan chased Janet around the playground, yelling for her to give back his gum. Janet laughed and kept running, eventually out of sight. Joan stomped back to his friends, still mad but with a plan. He would get his gum back, no matter what. | Joan got mad when Janet stole his gum. Not wanting to be outdone, Joan decided to steal something back. **Her favorite toy car broke loose.** Joan snatched Janet's favorite stuffed animal and ran away with it. Janet chased after Joan, but she was too late, as the toy car had already reached its destination. | <AFFGEN > story does not make sense. |

Table 9: Error analysis

| ID | Vanilla ChatGPT | Expanded AFFGEN story |
|---|---|---|
| 1 | Melissa invited Mark to the park for a picnic. It was a sunny Saturday, perfect for their long-awaited rendezvous. Melissa had spent hours preparing sandwiches, fruit salads, and homemade lemonade. As they spread out the checkered blanket, Mark noticed a curious twinkle in Melissa's eye. "What's the surprise?" he asked. With a mischievous grin, she pulled out a vintage camera. "I thought we could capture memories today." They laughed, took goofy selfies, and snapped candid shots of each other. The sun cast a warm glow, creating a picturesque backdrop. Later, Melissa reached into her bag, revealing a small book. It was a scrapbook, filled with their past adventures together. Tears welled in Mark's eyes as he flipped through the pages, reminiscing about their journey as friends turned soulmates. As the sun set, they stargazed, cherishing the memories they had created that day and adding another chapter to their beautiful story. | Melissa had planned the perfect picnic for Mark at their favorite park. The day started off sunny, but as they laid out their blanket and unpacked their delicious spread, ominous clouds gathered overhead. Suddenly, a deafening clap of thunder echoed through the park, sending them scrambling for cover. Huddled under a nearby tree, Mark and Melissa exchanged nervous laughter. As the rain poured down around them, they found themselves sharing stories, dreams, and even a few stolen kisses. Time seemed to stand still as the storm raged on. When the last raindrop fell and the sun emerged from behind the clouds, Mark and Melissa emerged from their shelter, hand in hand. The park glistened with fresh rain, and the sky painted itself in shades of pink and orange as the sun began its descent. They sat down on their damp blanket, feeling closer than ever, and savored their makeshift romantic picnic as they watched the sun set over the park. It turned out to be an unforgettable day, filled with unexpected magic. |
| 2 | Ella wrote her grandma a letter, her small hand gripping the pencil tightly. She missed her visits to Grandma's cozy cottage in the woods, where they would bake cookies and read stories by the crackling fireplace. In her letter, Ella poured her heart out. She shared her adventures, like the time she found a rainbow-colored caterpillar or built a secret fort in the backyard. She also drew pictures of her favorite memories, hoping Grandma could see them through the words. Days turned into weeks, and Ella anxiously checked the mailbox. Then, one sunny morning, a letter arrived. Excitement danced in her eyes as she tore it open. Inside was a heartfelt response, filled with love and wisdom, reminding Ella of the magical moments they had shared. With Grandma's letter in her hand, Ella felt like she was holding a piece of their cherished cottage. She knew that no distance could break the bond they shared, and that their love would continue to grow through letters, just like the flowers in Grandma's garden. | Ella, with tears in her eyes, carefully penned a heartfelt letter to her beloved grandma, sharing her recent adventures and school triumphs. Weeks passed, and to her surprise, a letter from her grandma arrived, the envelope trembling in her hands. Ella eagerly unfolded the letter, only to be met with heart-wrenching news: her favorite aunt had succumbed to an unexpected battle with cancer. Tears streamed down Ella's cheeks as she absorbed the painful words. Memories of her aunt's infectious laughter and warm hugs flooded her mind. Despite the sorrow, she found solace in her grandma's words. Her aunt had lived a life filled with boundless love, adventures, and cherished moments. She had been a beacon of positivity, spreading happiness wherever she went. Ella vowed to honor her aunt's legacy by embracing life with the same zest and determination. Though her heart ached, she knew her aunt's spirit would continue to inspire and guide her through the ups and downs of life. |
| 3 | Jen had a big test tomorrow, and her nerves were getting the best of her. She had spent countless hours studying, poring over textbooks, and reviewing notes, but anxiety still gnawed at her. The weight of the impending test loomed over her like a dark cloud. She tried to calm herself with deep breaths and a reassuring pep talk. "You've prepared as best as you can, Jen," she whispered to herself. "You know the material. You've got this." Jen's friends, knowing the stress she was under, decided to surprise her. They showed up at her doorstep with a homemade dinner, colorful sticky notes of encouragement, and their unwavering support. Together, they created a cozy study space filled with laughter and motivation. As the night wore on, Jen's anxiety began to subside. With her friends by her side, she felt a renewed sense of confidence and determination. She knew that no matter what the test held, she had the support of some incredible friends who believed in her. | Jen had a big test tomorrow, and anxiety gnawed at her as she burned the midnight oil, pouring over textbooks and notes. Hours passed, her eyelids heavy, but she still felt unprepared. Doubt clung to her like a stubborn shadow. Just as the clock struck midnight, her roommate, Sarah, burst into the room with a grin that could light up a stadium. Breathless and excited, she shared the unexpected news: the professor had decided to give everyone an extra day to study for the test. Jen's heart swelled with relief and gratitude. It was like a lifeline thrown to a drowning swimmer. Tears of joy welled up in her eyes as she hugged Sarah tightly. The universe had given her a second chance, and she wasn't going to waste it. She spent the next day revising her notes and feeling more confident with every passing hour. When she finally walked into the exam hall, she was armed not only with knowledge but also with the indomitable spirit of resilience that had seen her through the long night. |

Table 10: Comparison between vanilla ChatGPT stories and expanded AFFGEN's stories. We used ChatGPT for story expansion. we instructed the model with the following prompt "Expand the story to 150 words. The story should be interesting. <AFFGEN-generated plots>.". While narratives generated by the vanilla ChatGPT model tend to be straightforward and lack of dramatic flair, the narratives produced by the expanded AFFGEN-generated approach exhibit dramatic undertones, contributing to a more captivating reading experience.

