# OpenReview forum: "Affective and Dynamic Beam Search for Story Generation"
_EMNLP/2023/Conference — EMNLP 2023 Findings_

### Official Review · Reviewer_Y6wU · 2023-07-26

**Soundness:** 4

**Excitement:**

3: Ambivalent: It has merits (e.g., it reports state-of-the-art results, the idea is nice), but there are key weaknesses (e.g., it describes incremental work), and it can significantly benefit from another round of revision. However, I won't object to accepting it if my co-reviewers champion it.

**Paper Topic And Main Contributions:**

The author aims to generate interesting stories through twists and turns. They proposed AFFGEN(Controlled Affective Story Generator) to improve the interestingness of text by controlling the coherence of text and utilizing the emotional dimension of vocabulary. Dynamic beam size occasionally explores longer search beams, and affective rearrangement reorders candidate sentences based on arousal and value scores. Their automated ratings on the dataset showed that AFFGEN generated stories that were high in Arousal score but higher than baselines in arousal score. The manual assessment considers how well the generated stories are scored on six dimensions, including coherence, emotional engagement, empathy, interestingness, and overall preference. The results showed that AFFGEN achieved leading human evaluations most of the time.

**Reasons To Accept:**

The author believes that creativity is the deficiency of the current generation model, so the author directly chooses to fine-adjust GPT-2 and GPT-3 to build AFFGEN, and the results also show that AFFGEN is better than GPT-2 and GPT-3 in creativity, which provides a new inspiration for story generation.

**Reasons To Reject:**

The proposed AFFGEN model requires the exploration of different beam sizes and affective rearrangements. It's unclear whether adding these two technologies will outperform a well-fine-tuned model while keeping the system design simple.

**Reproducibility:**

4: Could mostly reproduce the results, but there may be some variation because of sample variance or minor variations in their interpretation of the protocol or method.

**Reviewer Confidence:**

3: Pretty sure, but there's a chance I missed something. Although I have a good feel for this area in general, I did not carefully check the paper's details, e.g., the math, experimental design, or novelty.

---

> ### Author Rebuttal · Authors · 2023-08-29
>
> Dear Reviewer,
>
> Thank you for your thoughtful review and constructive suggestions. We appreciate the time you have invested in evaluating our work. We address the concerns you raised below:
>
> **Regarding comparing with fine-tuned model**
> We compared with GPT-large which is fine-tuned on RocStories. And AFFGENl outperforms this well-finetuned GPT2-large, as demonstrated in Table 1. We want to highlight that fine-tuning a model beyond the setup discussed in our paper  is not trivial as it requires a dataset of interesting stories only, which does not exist. As highlighted by [Roemmele (2021)](https://roemmele.github.io/publications/human_computer_authoring.pdf), the notion of “interesting stories'' carries inherent subjectivity and is greatly influenced by context. Future work can look into creating resources of interesting stories but it extends beyond the purview of our current research.
>
> Our method is technically sophisticated but this is out of necessity. As pointed out by [Roemmele (2021)](https://roemmele.github.io/publications/human_computer_authoring.pdf), the notion of interesting stories is subjective and contextual. So simple fine-tuning is not enough for this task. Existing LMs can generate coherent texts but do not understand what "interestingness" means to humans.  Instead, there is a need to carefully design methods that can generate interesting stories without compromising coherence. AFFGEN meticulously balances text's interestingness (via affective reranking and large beams) and coherence (via switching to smaller beams).

---

### Official Review · Reviewer_pHn9 · 2023-08-02

**Typos Grammar Style And Presentation Improvements:** 1. Line 366, focus --> focuses
2. The…
**Soundness:** 4

**Excitement:**

3: Ambivalent: It has merits (e.g., it reports state-of-the-art results, the idea is nice), but there are key weaknesses (e.g., it describes incremental work), and it can significantly benefit from another round of revision. However, I won't object to accepting it if my co-reviewers champion it.

**Paper Topic And Main Contributions:**

The paper is dedicated to enhancing story generation, aiming to make narratives more captivating and emotionally engaging. The core concept revolves around introducing intriguing twists during sentence generation. To achieve this, the authors introduce the AFFGEN model, which can predict an appropriate position within sentences and then insert the intriguing twist. Additionally, the authors propose two strategies, namely Dynamic Beam Sizing and Affective Reranking, to select words during decoding that elevate the sentence's arousal and valence scores while reducing perplexity and make the sentence more interesting. The study provides a comparison of AFFGEN with the latest GPT2 and GPT3 language models, evaluating both automatically and through human assessment. The results demonstrate that AFFGEN outperforms these models, producing narratives that are more interesting and cohesive.

**Questions For The Authors:**

1. In Sec 4.1, how do you form D(p) from the Writing-Prompt dataset? How do you identify or quantify intriguing twists in a sentence?
2. Both the arousal and valence scores are based on a lexicon. Are there out-of-vocabulary cases during evaluation? If so, how do you deal with it?
3. What is a valence score? Maybe add some explanations
4. Line 387, in the prompt example for GPT3, where is the first sentence of the story?
5. Why only compare to GPT 2 & 3?  Are they the SOTA in story generations?

**Reasons To Accept:**

The paper proposes Dynamic Beam Sizing and Affective Reranking methods to make the generated sentences more interesting. The evaluation shows that the proposed method can improve the intrigue and coherence of a generated story.

----- After rebuttal -----

The paper introduces the Dynamic Beam Sizing and Affective Reranking techniques as a means to enhance the appeal of generated sentences. The evaluation demonstrates the effectiveness of these approaches in augmenting the intrigue while keeping the coherence of generated stories. Key contributions lie in its simplicity and adaptability. The method employs a count-based model to identify one twist sentence and subsequently inserts the intriguing twists. This infusion elevates the arousal and valence of the sentences and meanwhile introduces variations to the rest of the story. The inclusion of dynamic beam sizing further bolsters the method's adaptability to different scenarios.

**Reasons To Reject:**

1. Overall, the idea seems too complicated and not interesting. AFFGEN needs to train a model to identify the insertion position of the twist and uses an off-the-shelf model to predict trigger likelihood, but simply refer to the arousal and valence scores to define "interesting". Why not directly consider the arousal and valence scores over the whole sentence?
2. The evaluation results are not convincing.
    - The comparison only involved GPT 2 & 3 but no other SOTA methods.
    - Table 1 doesn't show significant gains from AFFGEN to the baselines.
3. Many technical details are missing from the paper. See `Questions For The Authors`.

----- After rebuttal -----

More efforts are needed in selecting better baseline results. If there are no other comparable methods, we could probably carefully prompt GPT 2 & 3 to make it a fair comparison.  For example, since the proposed method explicitly leverage the concepts of arousal and valence, we can also include them in the prompt sentence, something like "Generate a story that has high arousal and valence scores. The total length ....".

**Reproducibility:**

3: Could reproduce the results with some difficulty. The settings of parameters are underspecified or subjectively determined; the training/evaluation data are not widely available.

**Reviewer Confidence:**

4: Quite sure. I tried to check the important points carefully. It's unlikely, though conceivable, that I missed something that should affect my ratings.

---

> ### Author Rebuttal · Authors · 2023-08-29
>
> Dear Reviewer,
>
> Thank you for your thoughtful review and constructive suggestions. We appreciate the time you have invested in evaluating our work. We address the concerns you raised below:
>
> **1) Insertion Point of the Twist**
>
> AFFGEN does not employ an additional model to identify the twist's insertion point. As mentioned in section 4.1, we use a data-driven methodology to shape a distribution $D_{IT}$, which characterizes the typical positions of intriguing twists within sentences. At the inference stage, we draw from $D_{IT}​$ to determine the insertion point of the twist. This strategy ensures that our approach remains both uncomplicated and lightweight, circumventing any additional training overhead.
>
> **2) Complexity of Generating 'Interesting' Stories**
>
> On “too complicated”: Generating compelling narratives is intrinsically challenging. Our approach, driven by linguistic and literary criticism theories, aims to regulate arousal,valence and coherence through our tailored decoding techniques. This is a technically challenging task and so the proposed approach might seem “complicated”.
>
> On “not interesting”: We are not sure what the reviewer means by this. We look forward to further clarification on this.
>
> On definition of interesting: The term 'interesting' carries with it multiple dimensions. We used arousal and valence as one way of characterizing interestingness and show that this works. We recognize the potential for future endeavors to further dissect and model the multifaceted nature of 'interestingness’ but it is out of scope of the current paper.
>
> **3) Considering Arousal and Valence Over the Entire Sentence**
> We did conduct experiments that contemplated this very approach. However, our results showed that directly modulating valence and arousal across the entire sentence has less than 20% chance generating interesting stories. We observe that content diversity is also limited, especially on model’s choices on the main verb of the sentence. This motivates us to dive deep to token level to carefully balance between coherence and interestingness.
>
> **4) Including other SOTA methods**
>
> We would like to highlight that the generation of intriguing stories is still in an early stage. There are limited methods specifically tailored to this task, which has restricted the breadth of our comparative study. We strategically chose GPT models  as our baseline comparisons because they are widely recognized as the state-of-the-art (SOTA) story generators.  [Modeling Protagonist Emotions for Emotion-Aware Storytelling](https://aclanthology.org/2020.emnlp-main.426.pdf), [Choose Your Own Adventure: Paired Suggestions in Collaborative Writing for Evaluating Story Generation Models](https://aclanthology.org/2021.naacl-main.279.pdf), and [The Authenticity Gap in Human Evaluation](https://aclanthology.org/2022.emnlp-main.406/). Outperforming GPT-2, GPT-3, and ChatGPT, as AFFGEN has demonstrated, is indicative of the effectiveness and the potential of our approach. We welcome concrete suggestions for any other method that is a better fit for a baseline.
>
> **5) Clarification on Table 1**
>
> Table 1’s goal is not necessarily to showcase significant gains in coherence over baselines but rather to underscore that in our pursuit of enhancing story interestingness, we have not compromised on the coherence. In particular, Table 1 evaluates interestingness (via Arousal score) and coherence (via PPL, Uni and Rub). For interestingness, we can see from the table that AFFGEN’s improvement over the baselines is statistically significant. For coherence, AFFGEN performs at par with the baselines. Some baselines have better scores but they are not always statistically significant. This indicates that AFFGEN produces interesting but coherent stories. We will add more clarification about this.
>
> **6) Getting intriguing twist distribution from WritingPrompts dataset**
>
> As we pointed out in line 201, we followed the methodology described in [Wilmont and Keller 2020](https://doi.org/10.18653/v1/2020.acl-main.161)’s paper. They used a hierarchical language model that encodes stories and computes surprise and uncertainty reduction through entropy-driven modeling to identify the twist of the story.
>
> **7) OOV scenario for NRC-VAD lexicon group**
>
> We measure arousal and valence scores based on NRC VAD Lexicon, which is built through human ratings of valence, arousal, and dominance for more than 20,000 English words. VAD’s vocabulary size is quite large and it is has been the standard way to quantify valence and arousal of sentences. [A survey of textual emotion recognition and its challenges](https://ieeexplore.ieee.org/abstract/document/9330790/), [Semeval-2018 task 1: Affect in tweets](https://aclanthology.org/S18-1001/), and [Humor Knowledge Enriched Transformer for Understanding Multimodal Humor](https://ojs.aaai.org/index.php/AAAI/article/view/17534). Given the comprehensiveness and high coverage of the NRC VAD Lexicon, this knowledge source should be representative enough to handle most cases. On the contrary, there could always be rare out-of-vocabulary cases which we believe the impact would be minor.
>
> **8) Clarification on valence score**
>
> As referenced between lines 343 to 347, the valence score of a sequence of tokens is the cumulation of the valence scores of its individual tokens. Drawing from the NRC VAD lexicon, valence gauges the emotional sentiment of a word: a high score signifies positive sentiments like happiness or joy, while a low score indicates negative sentiments such as sadness or anger.
>
> **9) Regarding missing first sentence in GPT-3 prompt**
>
> We presented the complete prompt in Table 8, Appendix. In line 387, due to the space limit, we removed the place-holder for the first sentence of the story. However, since it is causing confusion, we will put it back.
>
> **10) Regarding SOTA in story generations**
>
> Yes, GPT-style models are currently SOTA in story generations as we discussed in `7)`
>
> Our method is technically sophisticated but this is out of necessity. As pointed out by [Roemmele (2021)](https://roemmele.github.io/publications/human_computer_authoring.pdf), the notion of interesting stories is subjective and contextual. So simple fine-tuning is not enough for this task (as also demonstrated by GPT-2 result in Table 1). Existing LMs can generate coherent texts but do not understand what "interestingness" means to humans.  Instead, there is a need to carefully design methods that can generate interesting stories without compromising coherence. AFFGEN meticulously balances text's interestingness (via affective reranking and large beams) and coherence (via switching to smaller beams).

---

### Official Review · Reviewer_ME8S · 2023-08-04

**Soundness:** 2

**Excitement:**

3: Ambivalent: It has merits (e.g., it reports state-of-the-art results, the idea is nice), but there are key weaknesses (e.g., it describes incremental work), and it can significantly benefit from another round of revision. However, I won't object to accepting it if my co-reviewers champion it.

**Paper Topic And Main Contributions:**

This paper introduces a new decoding algorithm to generate interesting stories. The algorithm works by generating most of the sentences using the beam search but generating a single sentence with an intriguing twist with the new decoding algorithm. Which sentence contains the intriguing twist is sampled from a prior distribution. The new decoding algorithm works by dynamically choosing a beam size and reranking the results with hand-crafted rules. The proposed algorithm is compared to GPT2, GPT3 (base model without applying the new decoding algorithm), and ChatGPT as baselines. The authors claim that both automatic evaluation and human evaluation show that the proposed algorithm can generate more interesting stories than the baselines. Ablation study shows that dynamic beam sizing helps improve the interestingness.

**Questions For The Authors:**

In Table 1, which language model is used for calculating PPL (for evaluating fluency)?

**Reasons To Accept:**

Improving the interestingness of the story is an interesting area of text generation but it is not well studied. The proposed algorithm is relatively simple: just generating one sentence containing an intriguing twist. The results show the method to be effective, significantly improving upon the base model with the standard decoding algorithm. The result that dynamic beam sizing improves interestingness compared to using a constant large beam size is somewhat surprising. Because the motivation to use dynamic beam size is that a larger beam size can generate more intriguing words but at the expense of coherence. The motivation for using dynamic beam size (as explained in Section 4.3) is to achieve a better trade-off between coherence and interestingness (as well as achieving better decoding efficiency).

**Reasons To Reject:**

The main weakness is that the proposed algorithm seems to be tailored to the five-sentence story dataset, which is the only test dataset. For example, many examples in the paper show that by generating the second sentence with an intriguing twist, the rest of the story becomes more interesting. This may be true for the five-sentence story, but it's hard to imagine this is the case for long stories. Also, the affective reranking contains many hand-crafted rules, which also seem to be tailored to the five-sentence story dataset.

Another weakness is that the comparison between GPT3 and ChatGPT seems to be unfair. While the authors build many hand-crafted rules  for affective reranking, the prompts for GPT3/ChatGPT are not well optimized. I played with ChatGPT and I can easily get more interesting stories by changing for prompt slightly from the prompt in Table 8. For example, by changing "Write a 5-sentence interesting short story" to "Write a 5-sentence interesting short story with a mystery".

I think the most interesting component is the dynamic beam sizing. While the ablation study shows it's better than a constant large beam size (in contrast to the original motivation), it does not give more insights into the underlying reason. So it is still not clear whether this is a generalizable method outside of the five-sentence dataset.

Finally, the clarity can be improved. It's not very clear how the contextual bandit model works.



**Reproducibility:**

2: Would be hard pressed to reproduce the results. The contribution depends on data that are simply not available outside the author's institution or consortium; not enough details are provided.

**Reviewer Confidence:**

3: Pretty sure, but there's a chance I missed something. Although I have a good feel for this area in general, I did not carefully check the paper's details, e.g., the math, experimental design, or novelty.

---

> ### Author Rebuttal · Authors · 2023-08-29
>
> Dear Reviewer,
>
> Thank you for your thoughtful review and constructive suggestions. We appreciate the time you have invested in evaluating our work. We address the concerns you raised below:
>
> **1) Regarding reviewer's comment on over-optimizing our approach toward 5-sentence short stories and its possible lack of generalizability to longer stories**, there are a couple of things we want to clarify:
>
> Story generation is a challenging task. One of the challenges is maintaining a  balance between interestingness and coherence. While coherence is not a big issue anymore for modern models for shorter stories, it is still a major challenge for longer stories. Therefore, we decided to work with short stories to focus our attention on interestingness which is a currently difficult problem.
>
> We argue that generating short, interesting stories can be a vital starting point in the broader quest to produce longer engaging texts. Since the community is still in the early stages of understanding what constitutes an interesting narrative,  this work can contribute valuable insights and methodologies for this purpose.However, this focus on short stories does not limit the potential application of our model to longer narratives.
>
> Inspired by literature criticism research, we also argue that longer stories could be created by expanding on the short stories that our model can generate. Here we would like to report additional experiment results: By prompting ChatGPT “Write a story based on the context we provide '', we expanded AFFGEN generated stories to long stories and compared with vanilla ChatGPT generated stories. The human evaluation shows that expanded-AFFGEN long stories have **40% win, 15% lose, 35%** tie rate compared with ChatGPT generated stories. We will include expert annotation for this experiment in the camera-ready version. By focusing on generating engaging short stories, this work lays the foundation for future research generating more complex, long-form content.
>
> **2) Regarding reviewer's comment on under-optimizing LLM’s prompt**, We want to clarify that
>
> **i)** AFFGEN is generalizable to different prompts: We recognize that diverse prompts to the story-infiller can have different effects on the generated content. We want to clarify that the strength of AFFGEN is providing intriguing twists regardless of base stories as the example shown below:
>
> `ChatGPT`: Tom is going to fish on Saturday. Beneath a rock, he finds a peculiar map. It leads to a forgotten cave. Inside, ancient markings puzzle him. A glint reveals a hidden compartment. A single question: "What lies beyond?" Determined, he started his adventure to unveil the truth.
>
> `AFFGEN`: Tom is going to fish on Saturday. He arrives at his favorite spot, waits, and casts line. <Two people start yelling loudly, threatening with violence.>. Perplexed, Tom notices their eyes flicker towards the trees. Hidden figures manipulating a bizarre scene? He decides to uncover the enigmatic pranksters.
>
> Note that the sentence wrapped in angle brackets is the intriguing twist generated by AFFGEN. We want to stress that ChatGPT's stories are indeed more coherent and it uses fancier words which has certainly impressed all users. However, there is more to storytelling than using fancy words. As the example illustrates, while ChatGPT gives a common treasure hunting story, AFFGEN writes a suspenseful story with the help of the intriguing twist. We want to highlight the generalizability of AFFGEN, which leverage words’ affective dimensions to promote text interestingness regardless of the prompt.
>
> **ii)** Specific prompts can indeed give you interesting stories but they get repetitive in spirit. We generated 10 stories using the prompt “Write a 5-sentence interesting short story with a mystery”. However, we found out that 7 out of 10 stories are talking about a secret being “unraveled” at the end. E.g. A photographer discovering a secret in a haunted lighthouse, a young woman discovering a hidden garden, and a man finding the hidden treasure… Our concern is that repetitive storylines would limit the scope of this research.
>
> As mentioned in line 385, we chose the prompt in the paper after experimentations without loss of generality. However, we do recognize that optimizing instruction could potentially improve the quality of generated stories. Future research could study how prompt engineering, alignment, and human-in-the-loop storytelling can empower the task of generating interesting stories, but is out of scope for the current paper.
>
> **3) Regarding reviewer’s question on interpreting ablation study**, We would like to better support our ablation study with the following examples:
>
> `AFFGEN_large_beam_size_60`: Ella wrote her grandma a letter. The big scary wolf attack scare hit hard.
>
> `AFFGEN_small_beam_size_10`: Ella wrote her grandma a letter. The letters would get lost.
>
> Given the example above, when we fix a large beam size, the generated text is interesting but not coherent. The reason behind the poor coherence is because during decoding, AFFGEN with a fixed large beam size chose a series of low-probability but high arousal tokens. While promoting the interestingness of the sentence, it hurt the story coherence and became not interesting. On the other hand, AFFGEN with a small fixed beam size generates content that is coherent but not as interesting. This shows the importance of not fixing the beam size and dynamically switching while the sentence is being generated.

---

### Meta-Review · Area_Chair_vQW4 · 2023-09-03

**Recommendation:** 3
**Confidence:** 3

**Metareview:**

This paper presents experiments in generating more or less interesting 5-sentences stories by manipulating the generator's beam search size dynamically and introducing a specific generation step for 'twist sentence' candidates and some lexicon-based reranking, which outperform GPT models in the automatic and human evaluation. Reviewers liked the task idea and some of the proposals in the paper, but criticized the very tailord nature of the task (exactly 5-sentence stories, in which for example a twist in sentence 2 seems most important), the lack of experiments using more effective prompts for the GPT comparisons, and the relatively weak results in Table 1 - as the rebuttal points out, the main positive result here hinges only on the arousal score, which is highly dependent on individual lexical items based on Mohammad 2018, and may have some connection with the lexicon-based reranking. Having read all of the rebuttals I am not completely convinced by the authors' argument about limitation to 5-sentence stories based on LLMs' difficulties in creating longer coherent stories - especially when heuristics are involved, a tailored approach will likely fare better when the outputs are more templatic in nature, and an attempt at a few-shot approach with examples of what the authors want from the model is also missing here. The latter point is especially important given the claim that LMs don't understand what humans find interesting - they cannot know what the authors are targeting exactly, unless we give them examples. At the same time, I agree that the experiments form an interesting first step in exploring this task and the nature of 'twists', so although reviewers were consistently on the fence with excitement scores of 3, if there is space, I would rather see this paper published than rejected. If it is not accepted to the main conference, I think it would make a good candidate for a Findings paper.

---

### Decision · Program_Chairs · 2023-10-07

**Decision:**

Accept-Findings

**Comment:**

This paper presents experiments in generating more or less interesting 5-sentences stories by manipulating the generator's beam search size dynamically and introducing a specific generation step for 'twist sentence' candidates and some lexicon-based reranking, which outperform GPT models in the automatic and human evaluation. Reviewers liked the task idea and some of the proposals in the paper, but criticized the very tailord nature of the task (exactly 5-sentence stories, in which for example a twist in sentence 2 seems most important), the lack of experiments using more effective prompts for the GPT comparisons, and the relatively weak results in Table 1 - as the rebuttal points out, the main positive result here hinges only on the arousal score, which is highly dependent on individual lexical items based on Mohammad 2018, and may have some connection with the lexicon-based reranking. Having read all of the rebuttals I am not completely convinced by the authors' argument about limitation to 5-sentence stories based on LLMs' difficulties in creating longer coherent stories - especially when heuristics are involved, a tailored approach will likely fare better when the outputs are more templatic in nature, and an attempt at a few-shot approach with examples of what the authors want from the model is also missing here. The latter point is especially important given the claim that LMs don't understand what humans find interesting - they cannot know what the authors are targeting exactly, unless we give them examples. At the same time, I agree that the experiments form an interesting first step in exploring this task and the nature of 'twists', so although reviewers were consistently on the fence with excitement scores of 3, if there is space, I would rather see this paper published than rejected. If it is not accepted to the main conference, I think it would make a good candidate for a Findings paper.